# When Sketches Diverge, Language Converges: A Universal Feature Anchor for Domain-Agnostic Human Reconstruction

## Abstract

When humans sketch the same pose, no two drawings are alike. Synthetic sketches exhibit algorithmic precision with clean edges and consistent strokes, while freehand sketches diverge wildly—each bearing the unique abstraction, style, and imperfections of its creator. This fundamental divergence has long challenged 3D human reconstruction systems, which struggle to bridge the chasm between these disparate visual domains. We present a paradigm shift: while sketches diverge, language converges. A pose described as "arms raised overhead" carries the same semantic meaning whether drawn by algorithm or artist. Building on this insight, we introduce a universal feature anchor—natural language—that remains constant across visual variations. Our framework leverages text descriptions to guide feature learning, creating domain-agnostic representations that transcend the synthetic-freehand divide. At the technical core lies our Text-based Body Pose Head (TBPH), featuring a novel gating mechanism where language-derived features dynamically reweight spatial regions of sketch features. This text-guided attention enables the model to focus on semantically meaningful pose indicators while suppressing domain-specific noise and stylistic artifacts. By augmenting 26,000 sketch-pose pairs with rich textual descriptions, we enable cross-modal supervision that teaches our model to see past surface differences to underlying pose semantics. Extensive experiments demonstrate our method's superiority: we achieve 139.86mm MPJPE on freehand sketches, a 4.5% improvement over the state-of-the-art TokenHMR, and further outperform it by 11.08% in zero-shot generalization on a newly collected dataset. More importantly, we show true domain-agnostic performance—our model trained on both domains exhibits minimal degradation when tested on highly abstract amateur sketches. This work establishes language as a powerful intermediary for visual domain adaptation, opening new avenues for robust cross-domain understanding in computer vision.

## 1 Introduction

Consider a person standing with arms raised overhead. Capture this pose in a photograph and process it through an edge detection algorithm—you'll get a precise skeleton of clean lines and perfect angles, every stroke consistent and predictable. Ask a hundred different people to sketch the same pose, and you'll receive a hundred unique interpretations—some confident and bold, others tentative and abstract, each filtered through individual perception and artistic style. Yet despite this visual chaos, something remarkable remains constant: the semantic meaning. Whether extracted by algorithm or drawn by hand, the pose can still be described with the same words: *standing with arms raised overhead.*

This observation illuminates a profound challenge in computer vision. Current 3D human reconstruction systems excel when fed synthetic sketches—those algorithmically generated drawings with their predictable strokes and consistent patterns. But hand these same systems a genuine human sketch, with all its irregularities and artistic liberties, and performance degrades dramatically (Wang et al., 2023; Yang et al., 2021). The domain gap between synthetic precision and human expression has proven stubbornly resistant to conventional approaches.

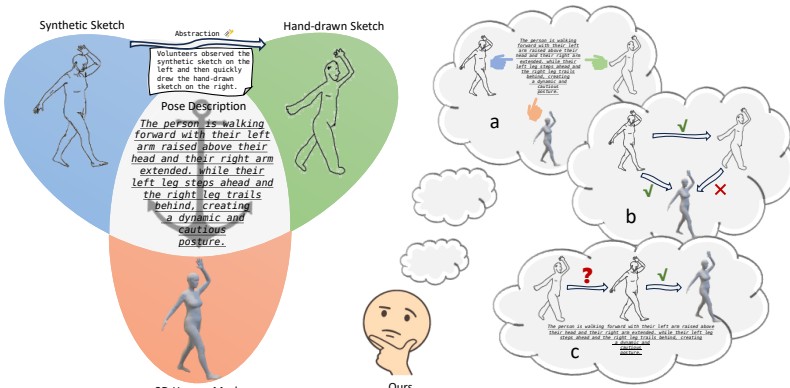

Figure 1: **Motivation.** This figure illustrates the motivation behind our approach. Both sketches depict the same pose: a person walking forward with their left arm raised above their head, right arm extended, left leg stepping ahead, and right leg trailing behind, creating a dynamic and cautious posture. The green sketch alone is insufficient to generate a high-quality 3D human mesh (red), whereas the blue (synthetic) sketch can. This work proposes using a pose description as a bridge to align hand-drawn sketches and synthetic sketches, thereby reducing the abstractness of hand-drawn input for 3D human mesh generation.

We argue that existing approaches have been constrained by their focus on reconciling visual representations that are fundamentally incompatible. As illustrated in Fig. 1, instead of attempting to force convergence between these inherently divergent visual domains, we propose leveraging a modality that naturally maintains consistency across different sketch styles: natural language. Language provides a domain-invariant representation—a pose described as *kneeling with hands on hips* carries identical semantic content regardless of whether the visual input originates from edge detection or freehand drawing. This linguistic invariance presents an unexplored opportunity for achieving domain-agnostic understanding in sketch-based reconstruction.

Previous attempts to bridge the synthetic-freehand divide have followed predictable patterns. Some approaches engineer intermediate representations (Yang et al., 2021), hoping skeletal keypoints might abstract away stylistic differences. Others pursue the data-driven path (Unlu et al., 2022; Wang et al., 2023), collecting ever-larger datasets to capture drawing variability. While these efforts yield incremental improvements, they fundamentally remain trapped within the visual domain, attempting to reconcile representations that are inherently irreconcilable.

The limitation becomes clear when we examine the state-of-the-art. Even TokenHMR (Dwivedi et al., 2024), among the best current methods, sees its performance plummet from 120.54mm to 146.37mm MPJPE when moving from synthetic to freehand sketches—a 21% degradation that reveals the fragility of purely visual approaches. This performance cliff isn't just a technical curiosity; it represents a fundamental barrier to real-world deployment where users naturally draw with human imperfection.

We propose a paradigm shift. Rather than viewing language as merely supplementary information, we position it as a *universal feature anchor*—a stable reference point that guides learning across visual domains. Natural language descriptions don't just label poses; they provide domain-invariant supervision that teaches models to see past surface variations to underlying semantic structure. Our framework, *UniAnchor*, operationalizes this insight through a novel Text-based Body Pose Head (TBPH) that fundamentally reimagines how visual and linguistic modalities interact. Unlike conventional attention mechanisms that compute similarities between modalities, our approach uses language-derived features to directly gate and modulate visual processing, dynamically highlighting semantically relevant regions while suppressing domain-specific noise. UniAnchor achieves 139.86mm MPJPE on freehand sketches—not just a 9.7% improvement over the previous best, but evidence of genuine domain-agnostic learning. More tellingly, when tested on highly abstract amateur sketches that would confound traditional systems, our method maintains robust performance, successfully reconstructing coherent 3D poses from inputs that barely resemble human forms. By establishing language as a bridge between divergent visual domains, this work opens new directions

for robust cross-domain understanding. The implications extend beyond sketch-based reconstruction to any scenario where visual appearance varies but semantic content remains constant—a common challenge across computer vision.

In summary, this work makes four key contributions:

- We identify language as a universal feature anchor that remains invariant across visual domains, providing stable supervision for cross-domain learning.
- Our Text-based Body Pose Head (TBPH) introduces a gating mechanism where semantic features directly modulate visual processing, achieving true domain-agnostic representations.
- We enrich 26,000 sketch-pose pairs with natural language descriptions, creating the first truly multi-modal resource for sketch-based reconstruction.
- Comprehensive experiments demonstrate not just quantitative improvements, but qualitative robustness to extreme abstraction and artistic variation.

## 2 RELATED WORK

**Text-Driven 3D Human Modeling.** The intersection of language and 3D human modeling is a rapidly evolving field. Early works focused on generating pose sequences from text Lucas et al. (2022); Zhang et al. (2022); Petrovich et al. (2021), while recent studies have expanded to modeling specific attributes like facial expressions Hou et al. (2022); Hwang et al. (2023); Zhang et al. (2024); Sun et al. (2022); Jiang et al. (2022) and clothing He et al. (2024); Huang et al. (2024); Youwang et al. (2022); Dong et al. (2024); Liu et al. (2024); Srivastava et al. (2024).

More advanced methods now enable sophisticated interactions, including text-based pose editing and correction Delmas et al. (2023); Kim et al. (2021). For instance, PoseFix Delmas et al. (2023) introduced paired data for pose modification via textual feedback. Concurrently, large multimodal models (LMMs) are being leveraged by methods like ChatPose Feng et al. (2023) for semantic and world-knowledge reasoning, while architectures like PoseEmbroider Delmas et al. (2025) integrate image, text, and 3D modalities for more fine-grained control.

However, key limitations persist. Text-only generation often lacks the necessary visual grounding to meet user specifications Delmas et al. (2022). Furthermore, approaches that do incorporate visual data, such as ChatPose Feng et al. (2023), are constrained by the known weakness of current LMMs in interpreting the abstract and nuanced details of freehand sketches.

**Vision-Based 3D Human Reconstruction.** Despite the success of 2D sketch-to-image synthesis Wu et al. (2023); Qu et al. (2024), 3D reconstruction is significantly harder due to the lack of depth information. Image-based 3D human reconstruction is typically divided into two main paradigms. The first approach regresses mesh vertices directly Moon & Lee (2020); Choi et al. (2020); Lin et al. (2021); Cho et al. (2022); Zhang et al. (2023). While excelling at capturing fine surface details, these methods require large datasets, struggle with occlusions, and are computationally expensive. The second category employs parametric human models like SMPL Loper et al. (2023); Pavlakos et al. (2019); Anguelov et al. (2005), offering greater anatomical plausibility and efficiency but with detail limited by the template's expressiveness Zanfir et al. (2021); Li et al. (2022); Zheng et al. (2023); Xuan et al. (2024); Shen et al. (2024); Su et al. (2025).

Regardless of the representation, a core challenge is inferring 3D structure from a 2D image. Landmark methods have progressively advanced this task. HMR Kanazawa et al. (2018) pioneered end-to-end regression, with subsequent work incorporating Graph Convolutional Networks (CMR Kolotouros et al. (2019b)), iterative optimization (SPIN Kolotouros et al. (2019a)), and multi-level attention (MAED Wan et al. (2021)). More recently, HMR 2.0 Goel et al. (2023) has demonstrated the power of pure Vision Transformer architectures.

A new paradigm has emerged using Vector Quantized Variational Autoencoders (VQ-VAE) to reformulate regression as a classification task over a learned codebook Fiche et al. (2025); Dwivedi et al. (2024). This includes methods that predict vertices (VQ-HPS Fiche et al. (2024)) and those that decode SMPL parameters (TokenHMR Dwivedi et al. (2024)). However, these vision-only approaches

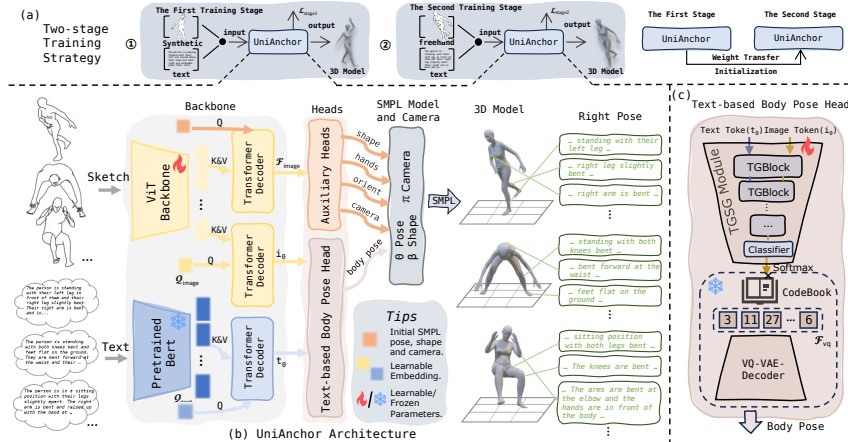

Figure 2: **(a) Two-stage Training Strategy. (b) UniAnchor Architecture:** Our framework processes sketch and text inputs through specialized encoders—a Vision Transformer for visual features and pre-trained BERT for semantic features. Image tokens flow through Transformer decoders and auxiliary parameter heads to estimate SMPL orientation ($P_{orient}$), hand parameters ($P_{hands}$), and camera parameters ($\pi$). **(c) Text-based Body Pose Head:** Both image and text tokens converge to predict body pose parameters ($P_{body}$), with language serving as a universal anchor for domain-agnostic learning. The complete SMPL parameters generate the final 3D human mesh.

falter on abstract freehand sketches, highlighting a need for the auxiliary semantic guidance that our language-anchored model provides.

## 3 PROPOSED METHOD

### 3.1 PRELIMINARIES

**SMPL Parametric Model.** We adopt the SMPL model (Loper et al., 2023), which parameterizes human body geometry through pose parameters $\theta \in \mathbb{R}^{144}$ and shape parameters $\beta \in \mathbb{R}^{10}$, producing a 3D mesh $V \in \mathbb{R}^{6890 \times 3}$. UniAnchor takes a sketch image $I$ and corresponding text description $T$ as input, predicting parameters $\hat{\Theta} = [\hat{\theta}, \hat{\beta}]$ and camera parameters $\hat{\pi} \in \mathbb{R}^3$. The 3D joint positions $J_{3D}$ are derived through learned joint regression from the predicted mesh.

**Dataset Augmentation with Language.** We augment the Sketch3D dataset using PoseScript (Delmas et al., 2022), which converts SMPL pose parameters $\theta$ into semantically rich textual descriptions. This heuristic, threshold-based generation avoids data leakage by ensuring the text is a high-level semantic abstraction of the pose, not a simple numerical transformation. This augmentation provides the domain-invariant supervision crucial for our approach.

### 3.2 NETWORK ARCHITECTURE

Fig. 2 (b) presents the UniAnchor architecture, comprising dual encoders for sketch and text processing, three specialized Transformer decoder modules, and two distinct prediction heads. The Auxiliary Heads predict global orientation, hand poses, shape, and camera parameters, while our novel Text-based Body Pose Head predicts body pose parameters using language as a universal anchor.

**Dual-Modal Encoders.** Following the success of HMR 2.0 (Goel et al., 2023), we employ a Vision Transformer (ViT) (Dosovitskiy et al., 2020) as our sketch encoder, producing image tokens of dimension $\mathbb{R}^{192 \times 64}$. For text encoding, we utilize the specialized encoder from PoseScript (Delmas et al., 2022) based on DistilBERT, which has been extensively trained on pose-related text, yielding text tokens of dimension $\mathbb{R}^{160 \times 64}$. We maintain learnable ViT parameters while freezing the pretrained BERT weights to preserve its semantic understanding.

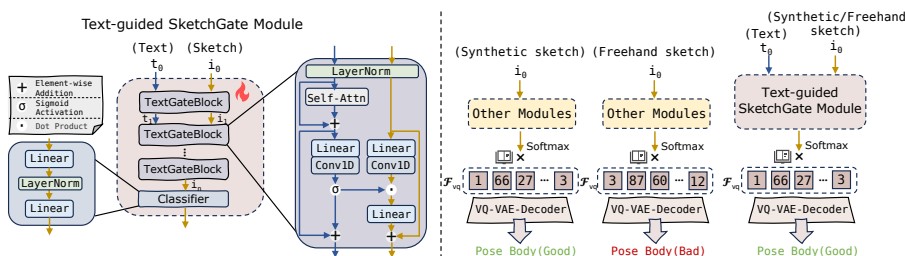

Figure 3: **Text-guided SketchGate Module and Domain Bridging Effect.** Left: The TGSG architecture comprises $n$ TextGateBlock modules (we use $n = 3$), where each block leverages text tokens to dynamically modulate image token outputs across spatial locations. Right: Visualization of how TBPH bridges domains. While conventional methods rely solely on image tokens (leading to divergent predictions for synthetic vs. freehand sketches), our text-guided approach reallocates token weights to align classification results across domains, effectively narrowing synthetic-freehand gap.

**Hierarchical Transformer Decoders.** Our architecture employs three Transformer decoders with distinct roles. The first decoder processes concatenated initial SMPL parameters $\Theta_{\text{init}}$ and camera parameters $\pi_{\text{init}}$ (forming a $1 \times 157$ dimensional query) with image tokens as keys and values, producing a 1024-dimensional feature vector $\mathcal{F}_{\text{image}}$. The second and third decoders utilize learnable queries $\mathcal{Q}_{\text{image}} \in \mathbb{R}^{160 \times 64}$ and $\mathcal{Q}_{\text{text}} \in \mathbb{R}^{160 \times 64}$ respectively, processing image and text tokens to generate refined representations $i_0$ and $t_0$, both with dimensions $160 \times 64$.

**Auxiliary Parameter Heads.** These MLP-based heads leverage $\mathcal{F}_{\text{image}}$ to predict auxiliary parameters: Specifically, four separate heads—the OrientHead, HandsHead, ShapeHead, and CameraHead—regress the global body orientation ($\hat{P}_{\text{orient}}$), hand pose ($\hat{P}_{\text{hands}}$), body shape ($\hat{\beta}$) and camera parameters ($\hat{\pi}$), respectively.

**Text-based Body Pose Head (TBPH).** Our key innovation, the TBPH, comprises two components: the trainable Text-guided SketchGate (TGSG) Module and a frozen VQ-VAE Decoder, as illustrated in Fig. 2 (c); this module (shown in Fig. 3 (left)) leverages text tokens to modulate weight distributions, aligning image token representations across synthetic and freehand domains. The resulting output is then fed into the VQ-VAE decoder to regress the body pose parameters.

The theoretical foundation rests on the observation that well-aligned text and image tokens should yield convergent probability distributions over pose space. As shown in Fig. 3 (right), when other modules are given a freehand sketch, they cannot obtain quantized features similar to those generated from synthetic sketches. However, by using the TGSG module to guide image token classification probabilities, we achieve similar quantized features across different sketch domains, effectively bridging the domain gap. Even for visually ambiguous sketches, textual information provides discriminative semantic cues that improve reconstruction quality.

**Text-guided SketchGate Module.** Unlike conventional gating mechanisms (Valanarasu et al., 2021; Cai & Wang, 2022; Yu & Wang, 2025; Hatamizadeh & Kautz, 2025) that derive gating weights solely from image features, we recognize that freehand sketches' inherent abstraction makes such approaches unreliable. Our TextGateBlock leverages cross-modal text features as stable, semantically rich signals for guiding weight allocation.

Each TextGateBlock processes input text token $t_{n-1}$ and image token $i_{n-1}$ through LayerNorm for cross-modal alignment. Text tokens undergo self-attention enhancement before both modalities pass through linear and 1D convolution layers, producing gating branches that interact via element-wise multiplication:

$$i_n = \text{Linear}(\sigma(t_{n-1}) \odot i_{n-1}) + i_{n-1}, \tag{1}$$

$$t_n = \sigma(t_{n-1}) + t_{n-1}, \tag{2}$$

where $\sigma(\cdot)$ denotes the Sigmoid activation and $\odot$ represents element-wise multiplication.

The final image tokens $\mathcal{F}_{\text{class}}$ are projected to CodeBook dimension $160 \times 2048$ via an MLP classifier, yielding category probabilities:

$$\mathcal{C}_{\text{prob}} = \text{Softmax}(\text{Classifier}(\mathcal{F}_{\text{class}})). \tag{3}$$

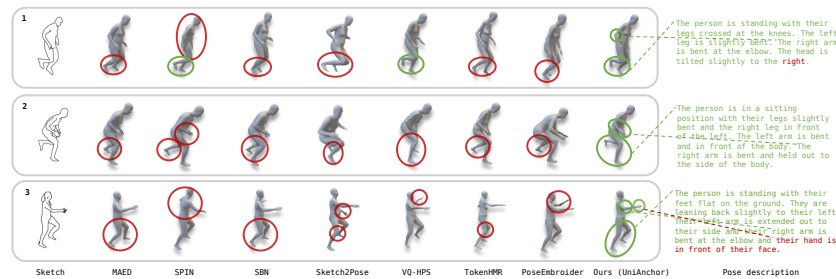

Figure 4: Visualization comparisons on the Sketch3D dataset. Red circles indicate errors in pose regions, green circles highlight successfully reconstructed poses. Red-highlighted text denotes discrepancies between the pose description and the sketch. Conversely, green-highlighted text signifies a successful match with the pose in the sketch. Finally, green lines represent actions consistent with the pose description, while red lines indicate inconsistencies.

**VQ-VAE Decoder.** Adopting the decoder from TokenHMR (Dwivedi et al., 2024), we leverage discrete vector quantization to align feature distributions across domains using discrete vector quantization. Our approach maps incoming features to a pre-trained codebook, and the resulting quantized features ($\mathcal{F}_{\text{vq}}$) are then decoded into the final body pose parameters ($\hat{P}_{\text{body}}$). The complete pose parameters are assembled as:

$$\hat{\theta} = \hat{P}_{\text{orient}} \oplus \hat{P}_{\text{body}} \oplus \hat{P}_{\text{hands}}, \quad \hat{\theta} \in \mathbb{R}^{144}, \tag{4}$$

generating the final 3D mesh through SMPL and deriving 2D projections via $\Pi$.

## 3.3 LEARNING STRATEGY

Following (Wang et al., 2023), as illustrated in Fig. 2 (a), we employ a two-stage training strategy. Stage one trains on synthetic sketches with comprehensive supervision:

$$\mathcal{L}_{\text{stage1}} = \mathcal{L}_{\text{vertices}} + \mathcal{L}_{J_{3\text{D}}} + \mathcal{L}_{J_{2\text{D}}} + \mathcal{L}_\theta + \mathcal{L}_\beta + \mathcal{L}_{\text{sim}}, \tag{5}$$

where $\mathcal{L}_{\text{vertices}} = \|\hat{V} - V^*\|_1$ is the mesh vertex loss, joint losses are $\mathcal{L}_{J_{3\text{D}}} = \|\hat{J}_{3\text{D}} - J_{3\text{D}}^*\|_2^2$ and $\mathcal{L}_{J_{2\text{D}}} = \|\hat{J}_{2\text{D}} - J_{2\text{D}}^*\|_2^2$, and parameter losses are $\mathcal{L}_\theta = \|\hat{\theta} - \theta^*\|_2^2$ and $\mathcal{L}_\beta = \|\hat{\beta} - \beta^*\|_2^2$.

Crucially, we incorporate a contrastive loss to align sketch and text features:

$$\mathcal{L}_{\text{similarity}} = \text{InfoNCE}(\bar{\mathcal{Q}}_{\text{image}}, \bar{\mathcal{Q}}_{\text{text}}), \tag{6}$$

where $\bar{\mathcal{Q}}_{\text{image}}$ and $\bar{\mathcal{Q}}_{\text{text}}$ are average-pooled query features. Stage two fine-tunes on freehand sketches with:

$$\mathcal{L}_{\text{stage2}} = \mathcal{L}_{J_{3\text{D}}} + \mathcal{L}_\theta + \mathcal{L}_{\text{similarity}}. \tag{7}$$

## 4 EXPERIMENTS

### 4.1 IMPLEMENTATION DETAILS

We employ a two-stage training strategy. First, we pre-train the model on synthetic sketches for 100 epochs with a learning rate of $2 \times 10^{-5}$. Then, using the checkpoint with the best validation performance, we fine-tune on freehand sketches for 10 epochs at a reduced learning rate of $5 \times 10^{-6}$. Both stages use the Adam optimizer, and all experiments were conducted on RTX 4090 GPUs.

### 4.2 EVALUATION PROTOCOL

**Metrics.** We adopt two widely-used metrics for 3D human pose evaluation, both measured in millimeters: **Mean Per Joint Position Error (MPJPE)** (Ionescu et al., 2013) quantifies the average

Table 1: Quantitative comparison with state-of-the-art methods on synthetic and freehand sketch datasets. ↓ indicates lower is better. **Bold** means the best result. * Indicates models trained or fine-tuned on our sketch dataset. † Indicates the model is not open-sourced. **PoseScript Text**: Uses original PoseScript descriptions. **Missing Text**: Uses empty text input. **Noisy Text**: Introduces noise by randomly swapping 'left' and 'right' keywords with 50% probability.

| Inference Input | Model | Synthetic sketch | | Freehand sketch | |
|---|---|---|---|---|---|
| | | MPJPE↓ | PA-MPJPE↓ | MPJPE↓ | PA-MPJPE↓ |
| Sketch | SPIN* | 133.55 | 84.54 | 185.09 | 99.98 |
| | MAED* | 125.73 | 81.10 | 176.79 | 97.24 |
| | Sketch2Pose† | 221.62 | 117.88 | 250.15 | 128.86 |
| | SBN* | 123.95 | 80.39 | 154.89 | 91.19 |
| | VQ-HPS* | 129.75 | 85.16 | 163.39 | 98.52 |
| | TokenHMR* | 120.54 | 83.55 | 146.37 | 92.06 |
| | **UniAnchor (Sketch & Missing Text)*** | **115.42** | **77.86** | **142.50** | **89.17** |
| Sketch + Text | PoseEmbroider* | 127.36 | 85.58 | 152.74 | 91.17 |
| | UniAnchor (Sketch & Noisy Text)* | 118.36 | 80.23 | 145.69 | 91.13 |
| | **UniAnchor (Sketch & PoseScript Text)*** | **112.99** | **76.23** | **139.86** | **86.68** |

Euclidean distance between predicted and ground-truth 3D joint positions:

$$E_{\text{MPJPE}} = \frac{1}{N_j} \sum_{i=1}^{N_j} \left\| \hat{J}_{\text{3D}}(i) - J_{\text{3D}}^*(i) \right\|_2, \tag{8}$$

where $\hat{J}_{\text{3D}}(i)$ and $J_{\text{3D}}^*(i)$ represent the predicted and ground-truth positions of the $i$-th joint respectively, with $N_j$ denoting the total number of joints. **Procrustes Aligned MPJPE (PA-MPJPE)** (Zhou et al., 2018) evaluates pose structure accuracy by first applying rigid Procrustes alignment to remove global positioning differences. This metric provides insight into the model's understanding of pose configuration independent of absolute position and scale.

**Dataset.** We use the Sketch3D dataset (Wang et al., 2023), which contains 26,000 poses, each paired with both a synthetic (Canny-based) and a freehand sketch. To enable cross-modal learning, we augment this data with text descriptions generated via PoseScript (Delmas et al., 2022).

## 4.3 COMPARISON OF STATE-OF-THE-ART METHODS

We compare our method with several state-of-the-art approaches, including SPIN (Kolotouros et al., 2019a), MAED (Wan et al., 2021), Sketch2Pose Brodt & Bessmeltsev (2022), SketchBodyNet (Wang et al., 2023), VQ-HPS (Fiche et al., 2024), PoseEmbroider Delmas et al. (2024) and TokenHMR (Dwivedi et al., 2024). Among these methods, SPIN, MAED, SketchBodyNet, Sketch2Pose, and TokenHMR regress the parameters of the SMPL/SMPL-X model to reconstruct 3D human meshes, while VQ-HPS directly predicts the 3D mesh vertices. Finally, 3D joint positions are obtained from the predicted vertices using a joint regressor and are then evaluated against the ground-truth joint annotations. Notably, PoseEmbroider shares the same input modality as Uni-Anchor, utilizing both sketches and textual descriptions. Additionally, since the training code for Sketch2Pose is not publicly available, we were unable to fine-tune it on the Sketch3D dataset. In contrast, all other competing methods were retrained on this dataset to ensure a fair comparison.

**Quantitative Results.** Crucially, It is important to clarify that all reported metrics for UniAnchor come from a single model trained under the standard protocol (using both Sketch and Ground-Truth Text). The variations—such as missing text, noisy text, or LLM-generated text—were introduced strictly during the inference phase to evaluate the model's robustness, without any modification to the training process.

Table 4.3 presents the comprehensive evaluation results. Our method achieves state-of-the-art performance across all metrics on both synthetic and freehand sketches. Several key observations emerge from these results: On synthetic sketches, UniAnchor achieves 112.99mm MPJPE and 76.23mm PA-MPJPE, representing improvements of 7.55mm and 7.32mm respectively over the previous best method (TokenHMR). These improvements demonstrate that semantic guidance enhances reconstruction even when visual features are clean and well-defined.

The performance gap becomes more pronounced on freehand sketches, where our method achieves 139.86mm MPJPE—a reduction of 6.51mm compared to TokenHMR. This 4.5% improvement is particularly significant given the challenging nature of freehand sketches. The PA-MPJPE improvement of 5.38mm (5.8% reduction) indicates that our method better understands pose structure, suggesting that textual descriptions help disambiguate visually ambiguous poses.

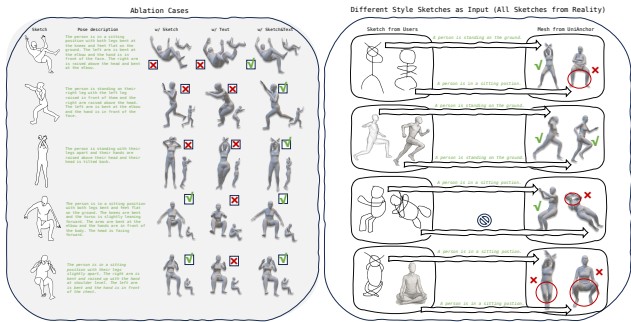

Figure 5: **Left:** Visualization results of our 3D human reconstruction method under different input modalities (Sketch, Pose description, or both). **w/ Sketch** denotes that the TGSG module uses only $i_0$ as input; **w/ Text** uses only $t_0$; and **w/ Sketch&Text** represents the full model. A green checkmark indicates close alignment with the sketch, while a red cross indicates misalignment. **Right:** Several real-world examples collected from a web application developed based on UniAnchor.

It is particularly worth noting that even under the "Noisy Text" and "Missing Text" settings, our method maintains a significant performance advantage over other state-of-the-art models. We attribute this robustness to the semantic alignment learned during training. The textual guidance effectively bridges the domain gap between synthetic and freehand sketches, thereby refining the intrinsic quality of the visual features. Consequently, the model retains competitive performance by relying solely on the enhanced visual encoder, even when explicit semantic information is absent or noisy.

Comparing across methods, SPIN exhibits the poorest performance with 185.09mm MPJPE on freehand sketches, highlighting its unsuitability for sketch-based inputs. MAED shows improvement over SPIN through its Transformer-based architecture but still struggles with the abstract nature of sketches. SketchBodyNet achieves reasonable results specifically designed for sketch inputs, while VQ-HPS shows inconsistent results across the two domains. PoseEmbroider attempts to construct a unified feature space for image and text modalities. However, it struggles to maintain this alignment across sketches with diverse styles due to the absence of explicit anchors.

Sketch2Pose serves as a zero-shot baseline in our evaluation. Although we could not fine-tune it on the Sketch3D dataset due to the unavailability of training code, it benefits from pre-training on a large-scale sketch corpus. In this setting, it delivers moderate performance (250.15mm MPJPE on freehand sketches), demonstrating basic generalization capabilities.

**Qualitative Analysis.** Figure 4 illustrates the 3D human reconstruction results of different methods on sketches depicting a variety of poses. The first sketch depicts a crossed-leg pose. The first three methods either fail to accurately reconstruct this action or misjudge the body orientation. Although VQ-HPS achieves a more accurate leg cross, it introduces noticeable artifacts in the arms. The second and third sketches depict a pose of standing on one leg with the other bent or extended. In this case, all competing methods generate a pose with the legs positioned close together, failing to capture the tension and separation implied by the sketch. In contrast, our method accurately perceives both the hand and leg configurations, successfully reconstructing the pose with raised arms and bent, separated legs, which closely matches the downward dynamic depicted in the sketch. Overall, our method demonstrates superior reconstruction quality across a wide range of complex poses, excelling in detail preservation and pose understanding, and significantly outperforming existing approaches.

## 4.4 ABLATION STUDIES

We conduct systematic ablation studies to validate our design choices and understand component contributions.

**Module Ablation.** Table 2 examines the contribution of each architectural component. Replacing the Text-based Body Pose Head (TBPH) with a simple MLP results in the most significant performance drop (6.66mm MPJPE increase on freehand sketches), confirming its critical role. When

Table 2: Ablation results of our method on the Sketch3D dataset, illustrating the impact of individual modules, modalities and loss.

| Ablations | Synthetic sketch | | Freehand sketch | |
|---|---|---|---|---|
| | MPJPE↓ | PA-MPJPE↓ | MPJPE↓ | PA-MPJPE↓ |
| **Modules** | | | | |
| TBPH-to-MLP | 117.80 | 79.53 | 146.52 | 89.04 |
| TGSG-to-SA | 115.82 | 76.95 | 143.33 | 88.07 |
| ImageGateBlock | 115.03 | 77.21 | 142.21 | 88.30 |
| TGB-to-CA | 116.56 | 79.84 | 147.27 | 89.49 |
| **Modalities** | | | | |
| w/ Sketch | 119.81 | 80.66 | 146.72 | 90.87 |
| w/ Text | 148.21 | 111.90 | 164.85 | 112.19 |
| **TextEncoder** | | | | |
| UnFrozen | 114.32 | 76.49 | 141.89 | 87.89 |
| **Caption Length** | | | | |
| Zero Sentence | 115.42 | 77.86 | 142.50 | 89.17 |
| One Sentence | 115.21 | 77.39 | 141.69 | 88.13 |
| Two Sentence | 114.09 | 76.51 | 140.24 | 87.86 |
| **Loss Function** | | | | |
| w/o InfoNCE | 113.54 | 76.86 | 141.38 | 87.88 |
| **Full Model** | **112.99** | **76.23** | **139.86** | **86.68** |

we substitute the Text-guided SketchGate (TGSG) with self-attention, performance degrades by 3.47mm, validating that our gating mechanism more effectively leverages semantic information than attention-based fusion. The ImageGateBlock variant, which uses only visual features for gating, underperforms by 2.35mm, demonstrating the value of cross-modal guidance. Notably, replacing our TextGateBlock with cross-attention causes a substantial 7.41mm degradation. This suggests that direct feature modulation through gating is more robust to domain shifts than similarity-based attention mechanisms.

**Modality Ablation.** Table 2 investigates the contribution of each input modality. Using only sketch input achieves reasonable performance (146.72mm MPJPE on freehand), confirming that visual information remains the primary signal. However, incorporating text reduces error by 6.86mm, demonstrating its value for disambiguation. Pure text-based reconstruction performs poorly (164.85mm MPJPE), as expected given the coarse nature of language descriptions. This refers to using text tokens to predict body pose parameters, while the remaining SMPL parameters are still predicted by image tokens. The significant performance gap between sketch-only (146.72mm) and text-only (164.85mm) approaches—18.13mm on freehand sketches—underscores that visual features provide essential geometric details that text cannot capture. Nevertheless, the synergistic combination outperforms both individual modalities, validating our multi-modal approach. Fig. 4.3 visualizes these findings, showing how text helps resolve ambiguities while sketches provide geometric constraints. In particular, the second example demonstrates how text-only prediction fails to capture correct hip positioning, while sketch-only prediction misplaces the arms, but the combined approach achieves accurate full-body reconstruction.

**TextEncoder Ablation.** We conducted an ablation study to determine whether to unfreeze the weights of the TextEncoder. As shown in Table 2, unfreezing the TextEncoder results in a marginal performance degradation. We attribute this to the distortion of the feature space among tokens in the unfrozen encoder during training. Although this accelerates loss convergence, it ultimately compromises the model's generalization capability.

**Caption Length Ablation.** We obtained text descriptions using PoseScript Delmas et al. (2022), where each image is typically associated with three sentences or fewer. To investigate the impact of text length on model performance, we segmented the descriptions by periods ("."). Table 2 presents the results ranging from zero to two input sentences. We observe that each additional accurate sentence yields an improvement of approximately 1mm in MPJPE. This demonstrates that accurate textual guidance significantly benefits model performance.

**Loss Function Ablation.** Table 2 shows that removing the InfoNCE loss increases MPJPE by 1.52mm on freehand sketches. While this improvement appears modest, it validates our hypothesis that explicit cross-modal alignment helps bridge the domain gap between synthetic and freehand sketches.

### 4.5 GENERALIZATION EXPERIMENT

To assess real-world applicability, we collected an additional 10,000 sketch-pose pairs following the Sketch3D protocol. We recruited over 30 new volunteers to draw freehand sketches, ensuring

Table 3: Generalization experiment results on our newly collected sketches. We test models with and without training on this dataset. **Bold** indicates the best result within each input modality group.

| Inference Input | Model | Synthetic sketch | | | | Freehand sketch | | | |
| | | Fine-tuned | | Zero-shot | | Fine-tuned | | Zero-shot | |
| | | MPJPE↓ | PA-MPJPE↓ | MPJPE↓ | PA-MPJPE↓ | MPJPE↓ | PA-MPJPE↓ | MPJPE↓ | PA-MPJPE↓ |
|---|---|---|---|---|---|---|---|---|---|
| Sketch | SPIN | 112.56 | 67.84 | — | — | 145.92 | 76.86 | — | — |
| | MAED | 98.45 | 61.57 | — | — | 130.61 | 69.33 | — | — |
| | Sketch2Pose | — | — | 209.27 | 101.52 | — | — | 257.50 | 117.58 |
| | SBN | 103.33 | 63.31 | 119.45 | 68.41 | 138.27 | 71.64 | 157.73 | 77.34 |
| | VQ-HPS | 78.52 | 46.71 | 113.08 | 62.68 | 125.74 | 66.77 | 155.17 | 75.26 |
| | TokenHMR | 73.34 | 45.41 | 113.77 | 61.28 | 105.89 | 59.01 | 153.48 | 76.70 |
| | **UniAnchor (Missing Text)** | **62.33** | **38.18** | **100.70** | **55.06** | **100.20** | **57.71** | **137.75** | **67.27** |
| Sketch + Text | PoseEmbroider | — | — | 109.82 | 60.75 | — | — | 141.74 | 69.38 |
| | UniAnchor (LLM Text) | 61.47 | 37.83 | 99.45 | 54.02 | 98.86 | 56.31 | 137.49 | 66.41 |
| | **UniAnchor (PoseScript Text)** | **58.35** | **36.10** | **98.84** | **52.43** | **96.85** | **53.58** | **136.47** | **65.83** |

diversity in artistic styles and skill levels. It is worth noting that, although this is a newly collected dataset, its sketch style is similar with that of Sketch3D Wang et al. (2023). The dataset was split 9:1 for training and testing. We structured the generalization experiments into two settings: fine-tuned and zero-shot. The fine-tuned setting investigates the model's adaptability to the new sketch domain, whereas the zero-shot setting directly employs weights trained on Sketch3D (with the exception of Sketch2Pose) to assess generalization capabilities on unseen sketches.

**Fine-tuned.** Table 3 demonstrates that UniAnchor maintains its performance advantage on unseen data. We achieve 96.85mm MPJPE on freehand sketches, outperforming TokenHMR by 9.04mm (8.5%). This larger improvement margin compared to the Sketch3D dataset suggests that language guidance becomes increasingly valuable as sketch diversity increases. The consistent superiority across both datasets confirms that our semantic anchoring approach enables robust generalization to varied artistic styles.

**Zero-shot.** Table 3 shows that our model achieves a reduction of 10.98mm in MPJPE on synthetic sketches and 5.27mm on freehand sketches compared to PoseEmbroider, demonstrating generalization capabilities superior to competing models. Notably, the model maintains excellent performance even with missing text descriptions or when using LLM-generated text (generated from Florence-2). We attribute this robustness to the guiding role of text, which aligns the feature space for sketches of varying styles but identical poses, thereby enhancing the Image Encoder's comprehension of diverse sketch representations.

### 4.6 LIMITATION AND FUTURE WORK

As shown in the left panel of Figure 4.3, coarse PoseScript descriptions (Delmas et al., 2022) currently serve only to fine-tune initial poses, proving insufficient for complex poses with severe joint overlapping. Real-world trials via our web application (Right) further revealed difficulties in reconstructing cross-legged postures and rotated arm crossings. To align with practical application scenarios, we employed simplified textual descriptions, which resulted in variable reconstruction performance. We attribute these failures to the limited pose diversity in our training set (∼20k images with high redundancy) and insufficient data augmentation, respectively. Furthermore, while text guidance currently enhances style comprehension, it does not yet play a dominant role in pose generation. To address these limitations, we plan to expand our dataset with diverse poses and precise annotations (manually or via LLMs) and design a robust architecture that more deeply integrates textual guidance to effectively resolve complex sketches.

## 5 CONCLUSION

In this paper, we presented UniAnchor, a novel framework that establishes natural language as a universal feature anchor for domain-agnostic 3D human reconstruction from sketches. By recognizing that while visual representations diverge across artistic styles, semantic descriptions remain consistent, we designed the Text-based Body Pose Head to leverage this invariance for robust cross-domain learning. Our comprehensive evaluation demonstrates state-of-the-art performance with a 4.5% gain over TokenHMR, extending to an 11.08% lead in zero-shot generalization to highly abstract amateur sketches. This work opens new directions for leveraging language as a bridge across visual domains, with implications extending beyond sketch-based reconstruction to broader challenges in cross-domain computer vision.

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

## A  APPENDIX

To evaluate perceptual quality, we conducted a user study with 12 graduate student volunteers (aged 20-25) assessing 3D reconstructions from 32 diverse sketches. Participants rated outputs on two 7-point Likert scales: (1) *faithfulness* - how accurately the pose matches the input sketch, and (2) *quality* - the visual quality and absence of artifacts.

The study employed a double-blind protocol where participants were unaware of which method generated each result. We collected 384 total evaluations (12 participants × 32 sketches). Fig. 7 presents the results, with a two-way ANOVA revealing statistically significant differences among methods for both faithfulness ($F_{(5,55)} = 33.01, p < 0.001$) and quality ($F_{(5,55)} = 30.27, p < 0.001$). UniAnchor achieved the highest scores with mean ratings of 5.68 ± 0.35 (faithfulness) and 5.86 ± 0.39 (quality), substantially outperforming all baselines. These results confirm that our quantitative improvements translate to perceptually superior reconstructions that better match user expectations.

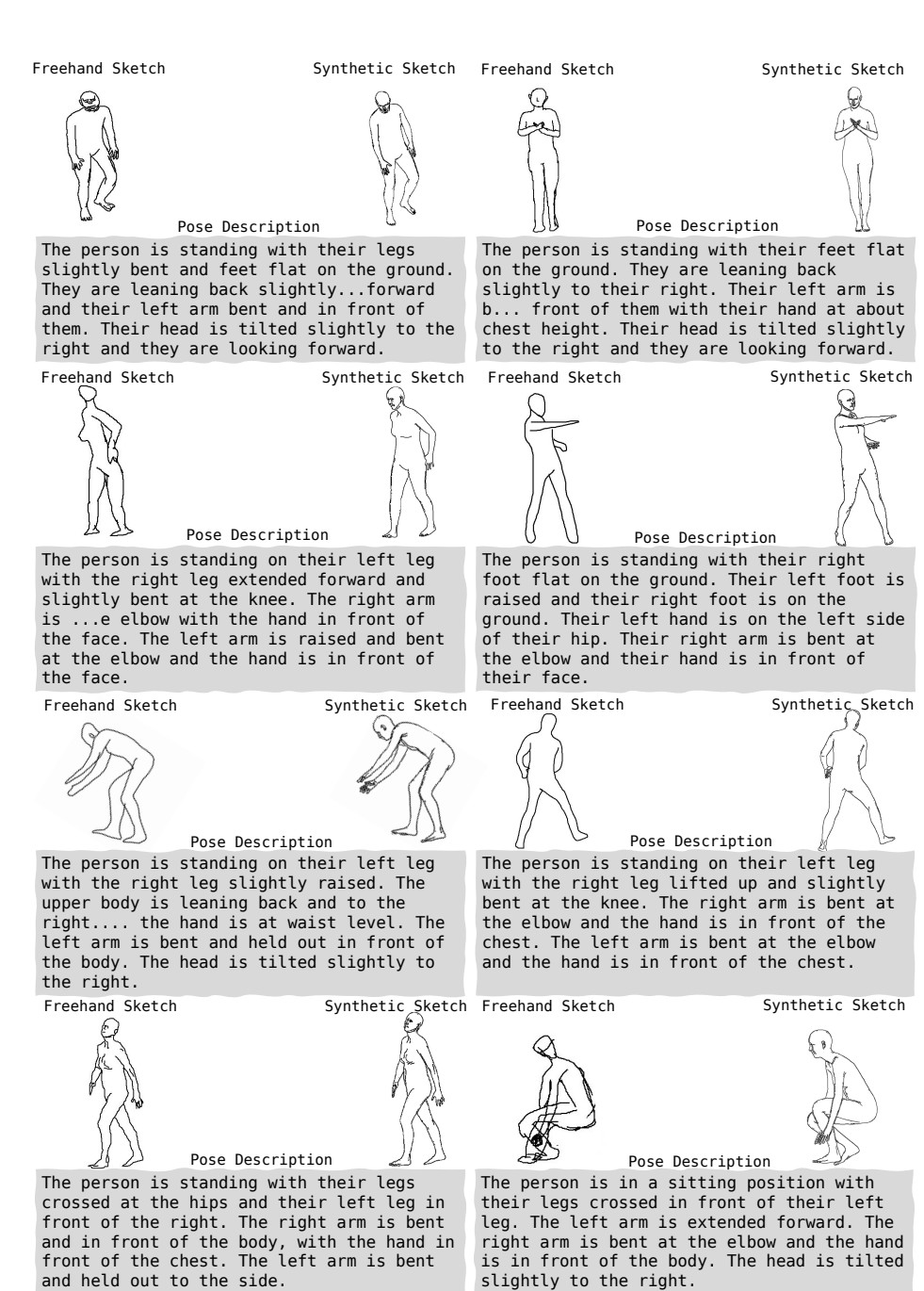

Figure 6: This figure shows the dataset of sketch-text pairs obtained using PoseScript (Delmas et al., 2022).

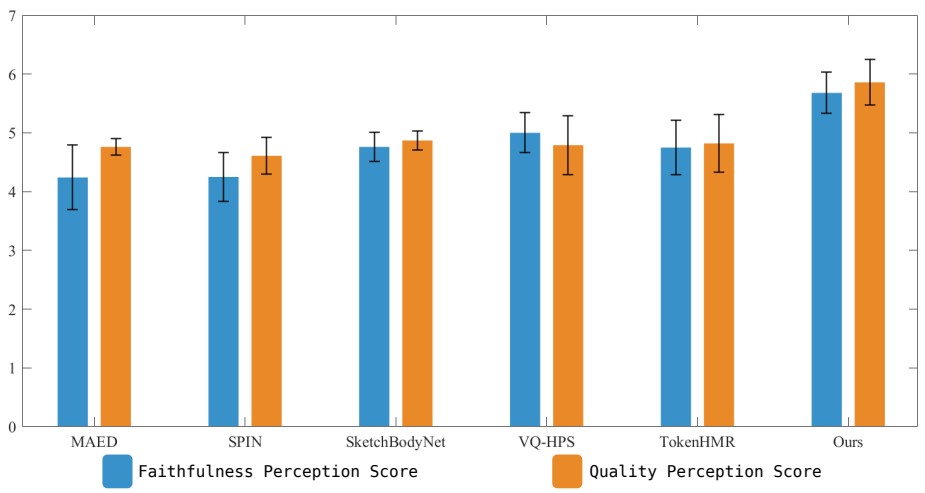

Figure 7: User study results showing faithfulness and quality perception scores. Blue bars represent faithfulness scores, orange bars indicate quality scores. Error bars show standard deviation.

