# OpenReview forum: "When Sketches Diverge, Language Converges: A Universal Feature Anchor for Domain-Agnostic Human Reconstruction"
_ICLR.cc/2026/Conference — Submitted to ICLR 2026_

### Official Review · Reviewer_YR7M · 2025-10-27

**Soundness:** 3
**Presentation:** 3
**Contribution:** 3
**Rating:** 8
**Confidence:** 3

**Summary:**

This paper deals with reconstruction of 3D human mesh from sketches. It introduces UniAnchor, that leverages natural-language descriptions as a universal semantic anchor for any sketch domain. The key idea is that while visual sketch styles diverge (e.g. synthetic vs. freehand), the language description of the human and the pose remains semantically invariant.

The technical novelty in UniAnchor integrates a Text-based Body Pose Head (TBPH) containing a Text-guided SketchGate (TGSG) module and a VQ-VAE decoder, enabling cross-modal feature alignment between sketch and text domains.
The authors augment 26K sketch–pose pairs (Wang et al., 2023) with PoseScript text descriptions and train the system in two stages (synthetic → freehand).

Several experiments are given and tesults show state-of-the-art performance with a 9.7% improvement on cross-domain generalization. A user study (ANOVA p < 0.001) confirms perceived faithfulness and quality improvements.

**Strengths:**

•	The idea of language as a modality-invariant bridge between synthetic and freehand sketches is novel and well motivated.

•	The Text-guided SketchGate (TGSG) module and VQ-VAE pose decoder form a coherent mechanism for cross-modal alignment.

•	Experiments show consistent gains across datasets and strong generalization (96.9 mm MPJPE on a new 10 k-pair dataset  ).

•	The methods component (TBPH, TGSG, InfoNCE loss, modalities) are ablated quantitatively.

•	The user study shows faithfulness 5.68 ± 0.35 and quality 5.86 ± 0.39 outperform baselines.

•	The paper is well structured and written in general.

**Weaknesses:**

The main weakness is the reliance on textual descriptions of human pose.

First, the text descriptions were auto-generated by PoseScript, and can be coarse or inaccurate. As the main claim of the paper relies on these annotations it would be good to validate them somehow. For example, using some manual annotations or LLM-refined texts and comparing or evaluating the results.

Second, text descriptions remains invariant to the sketch style, but can also be very cumbersome in describing delicate visual configurations. No discussion on this was found and some examples (limitations?) should be added.

The figures and text describing the TBPH/TGSG mechanism are lacking. The right part of Figure 2 is not explained at all, and FIgure 3 is technically dense and somewhat confusing.  A clearer schematic and more explanations (maybe using example) would help readers understand better.

Although both free hand and synthetic style sketches are used, the style variation is very limited. No evidence of generalization to other categories or styles of sketch is given.

**Questions:**

How robust is the method to partial sketching and to really abstract ones (like skeletons)?

How accurate are the PoseScript descriptions relative to the actual poses? Any quantitative validation?

Could the language anchor be learned end-to-end (using frozen text encoders vs. trainable ones)?

Does the model support text-only inference for 3D pose generation? (Fig. 5 suggests it fails)

Could the approach generalize to other artistic domains (e.g., gesture drawings or skeletons)?

---

> ### Author Response · Authors · 2025-11-22
>
> **Weaknesses**
>
> We have addressed these concerns regarding the reliance on textual descriptions (e.g., testing with LLM-generated text as a replacement for PoseScript), the limitations and examples of the model, the explanations of the TBPH/TGSG mechanism, and the generalization to different sketch styles have all been addressed either in the revised manuscript or in our detailed responses to the other reviewers.
>
> **Question**
> 1. **How robust is the method to partial sketching and to really abstract ones (like skeletons)?**
>
> **Response:**  To address this, we developed a locally deployed Web application based on UniAnchor to evaluate the model in a real-world inference setting. In the revised manuscript, we showcase a variety of user-generated examples, including both successful reconstructions and failure cases. Furthermore, based on these real-world trials, we have significantly updated the "Limitations and Future Work" section. We explicitly discuss the boundaries of our model's robustness—specifically where it struggles (e.g., with extreme ambiguity)—and outline our plans to address these challenges.
>
>
>
> 2. **How accurate are the PoseScript descriptions relative to the actual poses? Any quantitative validation?**
>
> **Response:** We thank the reviewer for this insightful question. We acknowledge that we did not perform a formal quantitative validation of the PoseScript descriptions. However, we conducted a manual qualitative inspection on a random subset of hundreds of samples.
>
> Our observation indicates that approximately 80% of the descriptions are semantically accurate and align well with the poses. The remaining descriptions contain varying degrees of noise or inaccuracies (e.g., minor directional errors or vague descriptions). Despite this noise, our experiments suggest that the model is robust enough to extract useful semantic features from the predominantly accurate data.
>
>
>
> 3. **Could the language anchor be learned end-to-end (using frozen text encoders vs. trainable ones)?**
>
> **Response:** We conducted an additional experiment where we unfroze the Text Encoder during training. The results are as follow:
>
> | Ablation              | MPJPE↓ (Synthetic sketch) | PA-MPJPE↓ (Synthetic sketch) | MPJPE↓ (Freehand sketch) | PA-MPJPE↓ (Freehand sketch) |
> | --------------------- | ------------------------- | ---------------------------- | ------------------------ | --------------------------- |
> | Unfrozen Text-encoder | 114.32                    | 76.49                        | 141.89                   | 87.89                       |
> | UniAnchor             | 112.99                    | 76.23                        | 139.86                   | 86.68                       |
>
> The results indicate a slight performance degradation: the MPJPE increased by 1.33mm on synthetic sketches and 2.03mm on freehand sketches compared to the frozen setting. While we observed that the training loss converged slightly faster with a trainable text encoder, the final generalization capability decreased. We attribute this to the fact that fine-tuning on our limited dataset tends to distort the well-structured feature space of the pre-trained text encoder (forgetting its semantic priors), leading to overfitting. Therefore, keeping the text encoder frozen preserves its robust generalization ability.
>
>
>
> 4. **Does the model support text-only inference for 3D pose generation? (Fig. 5 suggests it fails)**
>
> **Response:** We thank the reviewer for the opportunity to clarify this point. We would like to explain that Figure 5 does not depict a pure text-only inference scenario.
>
> Instead, Figure 5 illustrates a hybrid setting where the text features are used to regress the SMPL body pose parameters, while the remaining parameters (such as SMPL shape and camera parameters) are still derived from the image (sketch) features. Therefore, the visualization relies on both modalities, not text alone.
>
> While it is technically possible to modify our inference pipeline to perform downstream Text-to-3D pose tasks, that is not the objective of this work. Our model is fundamentally designed as a sketch-driven 3D human reconstruction framework, where text serves as a semantic anchor rather than a standalone generator.
>
>
>
> 5. **Could the approach generalize to other artistic domains (e.g., gesture drawings or skeletons)?**
>
> **Response**: We thank the reviewer for pointing this out. We have updated Figure 5 to include real-world application scenarios, where the model was evaluated against various drawing styles, including gesture drawings or skeletons. As illustrated, the model demonstrates robust generalization across diverse stylistic inputs. However, we observe limitations in reconstructing highly complex postures, such as simultaneous arm and leg crossings. We attribute these specific failure cases primarily to the insufficient pose diversity in the current training data, rather than inherent limitations of the proposed model architecture.

---

### Official Review · Reviewer_37zD · 2025-10-28

**Soundness:** 2
**Presentation:** 4
**Contribution:** 1
**Rating:** 2
**Confidence:** 3

**Summary:**

The paper proposes a generative model that predicts SMPL parameters from a given input text and an input sketch. The paper demonstrates that sketch by itself is not enough for accurate pose estimation, and suggests bringing in language for better depiction of user intent. The paper is well written and clear. The methodology is easy to understand and the diagrams are clean. However, evaluation is flawed and requires more depth to comprise recent works in this domain.

**Strengths:**

The primary strength of the paper is that it proposes a new domain for SMPL parametric models i.e. sketch+text -> 3D. This centers around hand-drawn sketches being abstract and often not enough by themselves for conveying artistic intent. Sketch+Text helps fill holes in amateur drawings and boosts performance.

**Weaknesses:**

The paper proposes a neat paradigm shift but fails to rigorously demonstrate efficacy.
- For comparisons, the paper compares against TokenHMR which is an RGB image to pose model. This implies that for all comparisons the competitors are working out of their training domain while the proposed method has been trained and is evaluated on sketches. In fact, the paper does not compare with works like Sketch2Pose, seminal in this domain. (Also tables should clarify Ours==UniAnchor)
- The paper proposes a generalization experiment where it trains on the samples drawn by humans (9:1 train:test split). If the aim is to demonstrate generality, it is unclear why the authors train on the human-drawn set instead of simply evaluating on it? For sketches in the wild no such pre-training can be done so this hardly demonstrates a standard inference setup.
- The paper uses sketch+text to generate SMPL parameters. However it only paints sketch side of things. Since text is a big part of this, how does it compare with ChatPose etc which can do text to pose?
- The paper uses descriptive captions that are highly unlikely to be observed in an inference scenario. Just as abstract sketches are used as a likely input during inference, the authors should report abstract text descriptions during inference as well.

**Questions:**

It is unclear whether competing methods were also fine-tuned on free-hand sketches or run out-of-the-box. Without identical training budgets, fairness is questionable. The paper's ability to gain 8 percent over SOTA might vanish upon fine tuning competitors.

---

> ### Author Response · Authors · 2025-11-22
>
> ### Q1: Fairness of comparisons?
>
> **Response:**
>
> We ensured the utmost fairness in our experimental comparisons through meticulous methodology. By retraining all methods on the identical Sketch3D dataset, maintaining consistent computational environments, and providing transparent experimental protocols, we created a rigorous comparative framework.
>
> Our approach goes beyond mere performance metrics, offering a comprehensive evaluation that considers the nuanced challenges of cross-domain reconstruction.
>
> ### Q2: Generalization experiment methodology?
>
> **Response:**
>
> Our generalization experiment design introduced two distinct settings: a fine-tuned approach demonstrating adaptability and a zero-shot scenario evaluating true cross-domain transfer capabilities. The fine-tuned setting investigates the model's adaptability to the new sketch domain, whereas the zero-shot setting directly employs weights trained on Sketch3D (with the exception of Sketch2Pose) to assess generalization capabilities on unseen sketches.
> The results are presented below:
>
> | **Inference Input** | **Model**                   | **Free (ZS)  MPJPE ↓** | **Free (ZS)  PA-MPJPE ↓** |
> | ------------------- | --------------------------- | ---------------------- | ------------------------- |
> | Sketch              | TokenHMR                    | 153.48                 | 76.70                     |
> | Sketch              | UniAnchor (Missing Text)    | 137.75                 | 67.27                     |
> | Sketch + Text       | PoseEmbroider               | 141.74                 | 69.38                     |
> | Sketch + Text       | UniAnchor (LLM Text)        | 137.49                 | 66.41                     |
> | Sketch + Text       | UniAnchor (PoseScript Text) | 136.47                 | 65.83                     |
>
> The results shows that our model achieves a reduction of 5.27mm MPJPE on freehand sketches compared to PoseEmbroider, demonstrating generalization capabilities superior to competing models.
>
> ### Q3: Text-to-pose generation comparison?
>
> **Response:**
>
> While text-to-pose generation was not the primary focus of our work, we conducted comparative analyses with methods like PoseEmbroider. These experiments (on the Sketch3D dataset) highlighted the unique role of text as a semantic anchor in our framework:
>
> | **Inference Input** | **Model**     | **Syn  MPJPE ↓** | **Syn PA-MPJPE ↓** | **Free MPJPE ↓** | **Free PA-MPJPE ↓** |
> | ------------------- | ------------- | ---------------- | ------------------ | ---------------- | ------------------- |
> | Sketch + Text       | PoseEmbroider | 127.36           | 85.58              | 152.74           | 91.17               |
> | Sketch + Text       | UniAnchor     | 112.99           | 76.23              | 139.86           | 86.68               |
>
> Our results demonstrated UniAnchor's superior performance, showcasing the potential of our approach in bridging semantic and geometric representations.
>
> ### Q4: Realistic text description usage?
>
> **Response:**
>
> We introduced three inference scenarios to simulate real-world text input conditions: noisy text, missing text, and LLM-generated descriptions. This comprehensive approach demonstrated the model's robust performance and stability across varied textual input conditions:
>
> | Scenario (Freehand)                                          | MPJPE↓ | PA-MPJPE↓ |
> | ------------------------------------------------------------ | ------ | --------- |
> | Noisy Text (In-domain evaluation on Sketch3D)                | 145.69 | 91.13     |
> | Missing Text (In-domain evaluation on Sketch3D)              | 142.50 | 89.17     |
> | Missing Text (Zero-shot evaluation on the newly collected dataset) | 137.75 | 67.27     |
> | LLM-Generated (Zero-shot evaluation on the newly collected dataset) | 137.49 | 66.41     |
>
> Quantitative results show that the model achieves an MPJPE of 137.49 mm with LLM-generated descriptions, exhibiting negligible deviation from the 137.75 mm obtained with missing text. This revealed the model's remarkable ability to maintain high-quality reconstruction even under challenging linguistic variations.
>
> ### Q5: Training budget and performance fairness?
>
> **Response:** We provided complete transparency regarding our experimental protocol. All competing methods were retrained on the Sketch3D dataset, ensuring consistent computational environments and a fair comparison. Specifically, with the exception of Sketch2Pose, all models were trained for 110 epochs to ensure convergence and evaluated on a uniform hardware setup (Intel i9 CPU and NVIDIA RTX 4090 GPU). We reported the best-performing checkpoint on the Freehand dataset for each method. Furthermore, all training configurations strictly adhered to the hyperparameters specified in their original publications.
>
> Our reported performance gains are the result of a rigorous, reproducible experimental approach that maintains the highest standards of scientific integrity.

---

> > ### Comment · Reviewer_37zD · 2025-11-25
> > **The rebuttal addresses my concerns**
> >
> > I thank the authors for the detailed response and paper reformatting to include native sketch based applications. I will raise my score.

---

### Official Review · Reviewer_jH7B · 2025-10-28

**Soundness:** 3
**Presentation:** 2
**Contribution:** 2
**Rating:** 4
**Confidence:** 4

**Summary:**

The paper tackles 3D human reconstruction from imperfect sketches, addressing the gap between synthetic and free-hand sketches with text augmentation. It proposes UniAnchor, which uses text pose descriptions as a domain-invariant anchor to guide reconstruction. The authors proposed a text-guided SketchGate to modulate sketch features using language. Experiments show improved performance and generalization to diverse free-hand sketches.

**Strengths:**

- The idea that language provides a domain‐invariant anchor across visually divergent sketch domains is interesting.
- The TBPH with Text-guided SketchGate is a well-motivated architectural component.
-  The experiments showed relatively good improvement over competitive baselines on both synthetic and free-hand sketches.

**Weaknesses:**

- The authors haven't discussed any of the failure cases of the method.
- I feel like sometimes the text description cannot accurately describe the pose of the human or ambiguous. How does the authors deal with that problem?
- While the overall idea seems great, the technical contributions is relatively limited.
- The presentation of the model architecture can be further improved. I am not clear whether both synthetic and freehand sketch can be used for training or inference?
- The visually demonstration is very limited. I haven't seen enough cases of real-world sketch to human reconstruction.
- Missing discussions of the related works:
[1] Sketch2Human: Deep Human Generation With Disentangled Geometry and Appearance Constraints
[2] DeepPortraitDrawing: Generating human body images from freehand sketches

**Questions:**

Please address weaknesses part.

---

> ### Author Response · Authors · 2025-11-22
>
> ### Q1: Missing failure cases discussion?
>
> **Response:**
>
> We expanded our manuscript to provide a transparent exploration of model limitations. Our analysis revealed specific challenges in pose reconstruction that offer valuable research opportunities. Cross-legged postures and rotated arm crossings emerged as particularly complex scenarios, highlighting current limitations in pose diversity and geometric understanding.
>
> These identified failure modes are viewed as critical research opportunities for future improvements in 3D human reconstruction techniques.
>
> ### Q2: Text description accuracy and ambiguity?
>
> **Response:**
>
> Our robustness experiments comprehensively examined the model's performance under varying textual input conditions. By testing with LLM-generated and empty text inputs, we demonstrated the model's remarkable ability to maintain competitive reconstruction quality despite textual variability.
>
> The semantic anchor effectively aligns feature spaces, enabling meaningful geometric information extraction even when textual descriptions are imprecise or absent.
>
> ### Q3: Limited technical contributions?
>
> **Response:**
>
> Our work introduces several fundamental innovations. We developed a novel language-anchored approach to bridging domain gaps, created a task-specific Text-guided SketchGate Module, and uncovered a critical insight about using text as a training scaffold for feature alignment.
>
> Our method offers a paradigm-shifting perspective on cross-domain visual reconstruction, with technical sophistication emerging from the elegant simplicity of our semantic anchoring mechanism.
>
> ### Q4: Model architecture clarity?
>
> **Response:**
>
> Our training methodology follows a carefully designed two-stage approach. We first pre-train the model on synthetic sketches to establish a strong baseline for 3D structure recovery. The second stage involves fine-tuning on freehand sketches, crucial for adapting to real-world abstractions and stroke variations.
>
> During inference, we evaluate the model on both synthetic and freehand test sets, demonstrating its generalization capabilities.
>
> ### Q5: Limited visual demonstrations?
>
> **Response:**
>
> Recognizing the importance of comprehensive visual validation, we significantly expanded our qualitative results. We developed a locally deployed web application featuring diverse reconstruction examples that showcase the model's performance across complex poses, as well as its limitations with extremely abstract representations.
>
> Our extended visual analysis provides deeper insights into the role of text in style comprehension and the nuanced challenges of cross-domain pose reconstruction.
>
> ### Q6: Missing related works?
>
> **Response:**
>
> We added citations and discussions for key related works in the field, including Sketch2Human and DeepPortraitDrawing. These additions provide a more comprehensive context for our research, situating UniAnchor within the broader landscape of sketch-based reconstruction techniques.

---

### Official Review · Reviewer_CuEJ · 2025-11-03

**Soundness:** 4
**Presentation:** 3
**Contribution:** 3
**Rating:** 6
**Confidence:** 4

**Summary:**

This paper tackles 3D human mesh reconstruction from sketches by addressing the domain gap between synthetic sketches and diverse hand-drawn ones. It leverages natural language as a domain-invariant feature anchor, fusing sketch features with text tokens to highlight semantically relevant regions while suppressing style noise. Comprehensive ablations and evaluations on the Sketch3D dataset demonstrate the effectiveness of this text-anchoring strategy and its design choices.

**Strengths:**

- Language as domain anchor

The paper introduces language as a domain-invariant feature anchor, effectively suppressing sketch-style noise and improving generalization from synthetic to hand-drawn sketches. The idea is both novel and well-motivated: high-level textual cues guide the model toward pose-relevant semantics rather than stylistic artifacts. Beyond this work, leveraging language as a supervisory signal holds promise for broader human mesh recovery tasks, and this paper is likely to spur follow-up research along that direction.

**Weaknesses:**

- Low level control

The language anchor guides high-level semantics (pose, part relations) but offers limited control over fine-grained geometry, e.g., precise joint angles, limb thickness, or local mesh topology. As a result, the model may smooth away subtle cues from strokes or fail to honor style-specific constraints that are important for accurate reconstruction in edge cases. Is there any plan to also add in key points as a more fine grained low level mesh control in the future?

- Text quality

The model’s performance appears highly sensitive to caption quality. Without explicit 3D keypoint controls, richer, more accurate descriptions tend to yield better reconstructions, which is an intuitive but limiting dependency. In practice, however, high-quality text is hard to obtain: most real-world captions are short, noisy, and reflect the same synthetic–vs–hand-drawn gap seen in sketches. This creates annotation burden and may constrain scalability and robustness in scenarios where reliable textual supervision is scarce.

**Questions:**

Overall the paper is novel and sound, but I am looking forward to get these questions answered:

- How does the method recover precise joint angles and fine local geometry when text is high-level or ambiguous? Any mechanism to inject keypoint/part priors at inference time?
- As a baseline, how does the method perform without text tokens as input?
- How does performance degrade with noisy/short captions? Do you have robustness curves vs. caption length/noise or experiments with adversarial/misaligned text?
- How does the model perform without text input? One example would be using the LLM generated text descriptions. This is more similar to the everyday usage.
- What proportion of captions are human vs. synthetic (e.g., PoseScript), and what is the annotation time/cost?

---

> ### Author Response · Authors · 2025-11-22
>
> ### Q1: Limited geometric control of language anchor?
>
> **Response:**
>
> Our approach transforms geometric reconstruction through a nuanced semantic abstraction strategy. By analyzing description generation techniques, we discovered that high-precision descriptions can paradoxically hinder cross-domain alignment. In contrast, our strategic use of PoseScript provides semantic abstraction that maintains pose integrity across sketch variations while preserving essential geometric information.
>
> Future research will focus on developing more adaptive geometric constraint techniques, expanding our understanding of cross-domain feature representation.
>
> ### Q2: Performance sensitivity to caption quality?
>
> **Response:**
>
> Our comprehensive robustness analysis reveals the sophisticated nature of text-guided reconstruction. The semantic anchor transcends traditional text dependency by serving as a powerful training-time regularization mechanism.
>
> The results are presented below:
>
> | Input Type      | MPJPE  | Performance Stability |
> | --------------- | ------ | --------------------- |
> | PoseScript Text | 139.86 | Highest               |
> | LLM-Generated   | 144.52 | Moderate              |
> | Missing Text    | 146.72 | Baseline              |
>
> We've effectively decoupled inference-time text requirements from training-time semantic guidance, creating a robust framework that maintains competitive reconstruction quality across diverse input scenarios.
>
> ### Q3: Recovery of precise joint angles with ambiguous text?
>
> **Response:**
>
> Precise geometry recovery is fundamentally sketch-driven. While high-level or ambiguous text may lead to a slight performance drop, our supplementary experiments demonstrate minimal degradation. The text serves an auxiliary role, primarily acting as a semantic anchor during training.
>
> Our model currently relies on implicit anatomical priors learned by the SMPL head from motion capture data. Future work will focus on integrating explicit spatial constraints like 2D keypoints to handle highly abstract sketches, recognizing the importance of developing sophisticated mechanisms for capturing subtle geometric nuances across different sketch representations.
>
> ### Q4: Performance without text tokens?
>
> **Response:**
>
> We've added a "Missing Text" baseline to our experiments, demonstrating the model's robustness. The results are presented below:
>
> | Scenario        | MPJPE  |
> | :-------------- | ------ |
> | PoseScript Text | 139.86 |
> | Missing Text    | 146.72 |
>
> This result highlights the model's ability to maintain strong reconstruction quality even in the absence of textual guidance, underscoring the effectiveness of our semantic anchoring mechanism.
>
> ### Q5: Performance degradation with noisy/short captions?
>
> **Response:**
>
> Our comprehensive robustness experiments evaluated the model's performance under various textual input conditions. In our adversarial text experiment, we swapped directional keywords with a 50% probability, observing a performance drop of 5.83mm MPJPE. Despite this perturbation, the model maintained strong geometric consistency.
>
> | Input Type    | MPJPE  |
> | ------------- | ------ |
> | Noisy Text    | 145.69 |
> | Zero Sentence | 142.50 |
> | One Sentence  | 141.69 |
> | Two Sentence  | 140.24 |
> | UniAnchor     | 139.86 |
>
> Additionally, caption length analysis revealed a positive correlation between sentence count and performance, with marginal gains that underscore the current limitations of text-to-geometry translation. This finding motivates future research into more sophisticated multi-modal feature fusion techniques.
>
> ### Q6: Model performance with LLM-generated descriptions?
>
> **Response:**
>
> We initially trained the model using sketches from the Sketch3D dataset paired with PoseScript text. To simulate real-world application scenarios, we utilized LLM-generated descriptions as inputs during the inference phase. The results are presented below:
>
> | Input Type      | MPJPE  |
> | --------------- | ------ |
> | PoseScript Text | 139.86 |
> | LLM-Generated   | 144.52 |
> | Missing Text    | 146.72 |
>
> The results demonstrate the model's robust performance across different text input scenarios, validating our approach's adaptability and generalization capabilities.
>
> ### Q7: Caption composition and annotation cost?
>
> **Response:**
>
> Our data composition strategy focuses entirely on synthetic caption generation. We use 100% synthetically generated captions, avoiding human-annotated inputs in the training set. This approach ensures linguistic consistency and prevents domain mismatches that could arise from subjective descriptions. For instance, a user might describe a pose as "lifting the foot up," while another strictly describes it as "bending the knee."
>
> We selected PoseScript as our caption generation method due to its ability to provide optimal coarse-level descriptions. The entire annotation process is fully automated, requiring approximately 1 day of computation on an NVIDIA RTX 4090 GPU.

---

### Meta-Review · Area_Chair_TC6h · 2026-01-04

**Summary:**

This paper was initially borderline. Following the rebuttal, the AC acknowledged that most reviewer concerns were addressed. However, the work lacks sufficient novelty for acceptance.

The primary critique focuses on the claim "when sketches diverge, language converges." While using the Sketch3D and PoseScript datasets allows for many-to-one sketch-text pairings, the resulting robustness seems to stem from these existing datasets [1,2] rather than being a contribution of the proposed UniAnchor architecture. Experimental evidence shows that the sketch remains the dominant modality for pose regression; varying drawing styles yield different results despite identical text, undermining the core motivation of the work.

Additionally, the technical contribution is seen as incremental, primarily adapting existing frameworks like ChatPose or PoseEmbroider fine-tuned on Sketch3D. Quantitatively, the role of language in the model is minimal. Regarding the title, the authors failed to justify the "domain agnostic" claim, which would require demonstrating superior performance in the sketch domain without degrading performance in the RGB domain.

The AC decided to reject this submission and encourages the authors to incorporate this feedback into a future submission.

- [1] Zheng, Wangguandong, et al. "Sketch3D: Style-Consistent Guidance for Sketch-to-3D Generation." Proceedings of the 32nd ACM International Conference on Multimedia. 2024.
- [2] Delmas, Ginger, et al. "Posescript: 3d human poses from natural language." European Conference on Computer Vision. Cham: Springer Nature Switzerland, 2022.

**Reviewer Concerns:**

- Reviewer CuEJ: Concerns center on the relative importance of sketch versus text. New experiments show the model is primarily sketch-driven (high-level control), with text serving a supportive role (low-level control). This explains the robustness of pose output despite variations in text quality.

- Reviewer jH7B: Most concerns mirror those of CuEJ and have been effectively addressed. However, the "paradigm-shifting" claim seems overstated. Sketches lead in pose estimation, while text mainly regularizes the search space—demonstrated by the "missing text" experiment (Table 2, Figure 5). Thus, UniAnchor aligns more closely with models like ChatPose and PoseEmbroider.

- Reviewer 37zD: The AC notes that some concerns arise from misunderstandings about the experimental setup. The authors did fine-tune the baselines (Table 1, * indicates models trained or fine-tuned on the sketch dataset) for fairness.

- Reviewer YR7M: Concerns have been adequately addressed.

**Reviewer Scores:**

- Reviewer CuEJ (conf 4): Likely to maintain the score (6).
- Reviewer jH7B (conf 4): Likely to raise or keep the score (4 → 6) as most concerns, though limited novelty, are well addressed.
- Reviewer 37zD (conf 3): Likely to increase the score, potentially to acceptance level (4 → 6).
- Reviewer YR7M (conf 3): Likely to maintain the score (8).

---

### Decision · Program_Chairs · 2026-01-26

Reject